# Melanin Induction Restores the Pathogenicity of *Gaeumannomyces graminis* var. *tritici* in Wheat Plants

**DOI:** 10.3390/jof9030350

**Published:** 2023-03-14

**Authors:** Camila Aranda, Isabel Méndez, Patricio Javier Barra, Luis Hernández-Montiel, Ana Fallard, Gonzalo Tortella, Evelyn Briones, Paola Durán

**Affiliations:** 1Programa de Doctorado en Ciencias de Recursos Naturales, Universidad de La Frontera, Temuco 4811230, Chile; 2Biocontrol Research Laboratory, Universidad de La Frontera, Temuco 4811230, Chile; 3Scientific and Technological Bioresource Nucleus, Universidad de La Frontera, Temuco 4811230, Chile; 4Nanotechnology and Microbial Biocontrol Group, Centro de Investigaciones Biológicas del Noroeste, Av. Politécnico Nacional 195, Col. Playa Palo de Santa Rita Sur, La Paz 23090, Mexico; 5Facultad de Ciencias Agropecuarias y Medioambiente, Departamento de Producción Agropecuaria, Universidad de La Frontera, Temuco 4811230, Chile

**Keywords:** take-all disease, melanin, UV, vis spectrum

## Abstract

One of the most challenging aspects of long-term research based on microorganisms is the maintenance of isolates under ex situ conditions, particularly the conservation of phytopathological characteristics. Our research group has worked for more than 10 years with *Gaumannomyces graminis* var. *tritici* (Ggt), the main biotic factor affecting wheat. In this sense we preserved the microorganisms in oil overlaid. However, several strains preserved for a long time lost their pathogenicity. These strains show white and non-infective mycelia. In this sense, we hypothesized that this is attributable to low melanin content. Melanin is a natural pigment mainly involved in UV protection, desiccation, salinity, oxidation, and fungal pathogenicity. Therefore, understanding the melanin role on Ggt pathogenicity is fundamental to developing melanin activation strategies under laboratory studies. In this study, we induce melanin activation by UV-A light chamber, 320 to 400 nm (T1) and temperature changes of 30 °C, 15 °C, and 20 °C (T2). Fungal pathogenicity was evaluated by determination of blackening roots and Ggt was quantified by real-time PCR in inoculated wheat plants. Results revealed that Ggt grown under UV-A (T1) conditions showed around 40% higher melanin level with a concomitant effect on root infection (98% of blackened roots) and 4-fold more Ggt genome copy number compared with the control (non-infective mycelia) being T1, a more inductor factor compared with T2. These findings would support the role of melanin in pathogenicity in darkly pigmented fungi such as Ggt and could serve as a basis for activating pathogenicity under laboratory conditions.

## 1. Introduction

Wheat (*Triticum aestivum* L.) production is a key component underpinning worldwide food, and it is the primary source of calories and protein to 30% and 60% of human nutrition, respectively [1,2]. In fact, wheat is considered as a key food to global food security [3,4] and together with maize is a major source of dietary energy, of essential proteins and micronutrients, and diverse non-nutrient bioactive food components [5].

Despite the effort of the scientific community to increase wheat productivity and sustainability in diverse soil and production conditions, wheat production is highly affected by the soil-borne pathogenic fungi *Gaeumannomyces graminis* (Sacc.) var. *tritici*, (Ggt), a member of the Magnaporthaceae family, which causes “take-all” disease. The hyaline hyphae of Ggt penetrate the root cortex and destroy the vascular tissue, which results in black necrotic lesions characteristic of the disease and in severe infections hindering nutrient and water transport from roots to shoots, which reduced wheat yields by around 50% [4,6,7,8,9]. Due to its detrimental impact on wheat production, our research group has focused on the development of biotechnological tools for Ggt biocontrol [3,4,10,11,12]. For this, it is crucial to maintain and preserve pure and pathogenic microorganisms under ex situ conditions.

Cryopreservation, lyophilization, and preservation under liquid nitrogen are commonly used for long term preservation of different groups of fungi, but unfortunately they are expensive and need special equipment [13]. For example, for ex situ conservation by cryopreservation, the tubes should be frozen in an ultracold freezer from −70 °C to −196 °C and periodically inoculated to evaluate fungal viability, purity, and phenotypic characteristics [14]. The method of preservation by lyophilization consists of drying at −50 °C and vacuum sublimation. However, not all fungi survive the process conditions, and storage in liquid nitrogen is more expensive than lyophilization because the liquid nitrogen should be replenished every few days [14,15]. Therefore, oil overlay is particularly useful for mycelial or nonsporulating forms, which cannot be freeze-dried or frozen successfully, and in small laboratory collection [16]. In this sense, during routinary laboratory work, numerous pure Ggt culture, maintained in the oil overlay method for more than 24 months, reduced their pathogenicity, evidenced by white mycelia and failure assays of fungal inoculation in wheat due to asymptomatic plants. This loss of pathogenicity in Ggt has also been reported earlier [17]. For example, approximately less than half of the 111 strains of Ggt lost their pathogenicity after 9 months of culture on potato-dextrose agar. Interestingly, these authors reported that Ggt strains lost pathogenicity at about the same rate, whether were transferred every 10 days at 24 °C or if the Ggt strains were stored (without transfer) at 12 °C or 24 °C without making transfers. Moreover, Ggt cultures were more likely to remain pathogenic when, first, they were passed through the host, i.e., the fungus was inoculated in plant roots, and then reisolated from the infected roots at approximately 1-month intervals beginning as soon as possible after the fungus was isolated from the infected plants from nature. Finally, the authors suggest that the fungus has a low frequency of nuclei lacking genes for virulence or carrying genes that suppress expression virulence, this due to the heterokaryotic characteristics of the fungus. However, this phenomenon has been reported not only for Ggt strains. Successive cultivation of *Fusarium oxysporum* f. sp. *niveum* on artificial media has been reported to cause degeneration of developmental phenotype, and a reduced virulence [18]. These authors reported that, due to degenerative changes after repeated cultivation, a loss of pathogen virulence-related factors of the early stage of infection process was observed. In this sense, it was observed that cell wall-degrading enzymes such as pectinase, xylanase, and cellulase activities decreased significantly, in addition to notoriously decreasing in the transcript levels of *fmk1*, *fgb1*, *pacC*, *xlnR*, *pl1*, *rho1*, *gas1*, *wc1*, and *fow1*, which are nine virulence-related genes.

Several studies highlighted the important role of melanin on fungal pathogenic expression [19,20,21]. In this sense, melanin has been postulated to contribute to fungal virulence by reducing the pathogen’s susceptibility and by influencing their host immune response to infection [22]. Melanin is a dark pigment with high molecular weight formed by oxidative polymerization of indole or phenolic compounds with high molecular weight, generally red, black, or brown found in plants, animals, and microorganisms [16]. Thus, it is a unique pigment with diverse functions that are found in all biological kingdoms providing defence against environmental stresses such as ultraviolet (UV) light, oxidizing agents, and ionizing radiation or “fungal armor” [23,24,25]. In fungi, melanin may be confined at the cell surface or released into the extracellular space, to differ between different species, and it is closely related with molecular interactions with chitin structures [24,26]. Therefore, any disruption in chitin metabolism can lead a low melanin content [27]. According to its chemical and physical features, melanin can be classified into eumelanin, pheomelanin, neuromelanin, allomelanin, and pyomelanin [21,23]. Allomelanin is the most common melanin in fungi, and Dihydroxynaphthalene (DHN) melanin is the most common type of allomelanin [28], with an important role in fungal pathogenicity. For example, *Magnaporthe oryzae* use melanized appressoria as the sole means of penetration of the host surface [29,30,31], where melanin limits cell wall permeability, allowing the accumulation of osmolytes and the buildup of enormous turgor pressures in the appressoria [20,31]. This process creates the forces necessary for the penetration pin to rupture the host cell cuticle and epidermal cell wall. Thus, the differentiation of melanized appressoria is a key step in the process of leaf infection by *M. oryzae* [32,33]. In the case of the ascomycete *Venturia inaequalis* fungi, melanin is not a prerequisite for the pathogenicity but increases the success of cuticle penetration and conidia release in apples [34]. Additionally, melanin increases pathogenicity in fungi that cause human disease, such as *Cryptococcus neoformans*, *Aspergillus fumigatus* and *Wangiella dermatitidis* [35].

In the case of Ggt, in addition to cellulolytic and pectinolytic enzymes that aid in the infection of host cells, the role of melanin in the hyphae is important to the formation of melanized appressoria-like structures called hyphopodia. While it is known that melanin plays a key role in the infectivity and pathogenicity of pathogenic fungi, it is still not entirely clear what are the factors that induce the accumulation of this molecule, although environmental stressors appear to play a central role. Therefore, our main objective was to evaluate the influence of ultraviolet light (UV-A) light and temperature on the accumulation of melanin and its consequent influence on wheat plant pathogenicity.

## 2. Materials and Methods

### 2.1. Inoculum Preparation and Melanin Fungal Pigment Activation

*Gaeumannomyces graminis* (accession number KY689233), previously isolated and identified by sequencing the ribosomal internal transcribed spacer 2 (ITS-2) region [11,23], was grown on PDA for 15 days at 25 °C (white colonies, control, or low-melanized colonies). The melanin inductor factor 1 (T1) consisted of growth under an ultraviolet light chamber (UV-A 320–400 nm) at room temperature also for 15 days. Melanin inductor factor 2 (T2) had temperature fluctuations every 5 days, starting at 30 °C, 15 °C and 20 °C (15 days in total), and then melanin was quantified [36].

### 2.2. Detection and Quantification of Melanin Fungal Pigment

Four fungal mycelial plugs (5 mm in diameter) obtained from the periphery of the growing colony of each treatment (control, T1, and T2) were transferred into 200 mL of potato-dextrose broth, pH 6.0, and incubated under agitation (125 rpm) in the dark at 25 °C for 3 weeks. After incubation, the cultures were centrifuged at 11,000 rpm for 15 min to harvest the fungal mycelia. The fungal mycelia were dried at 60 °C for 48 h, and thereafter it was chilled down in desiccators for 20 min before being weighed and kept at 4 °C in the darkness [37].

Extraction and purification of pigment from dried fungal biomass were performed as described previously with some modifications [38,39]. Fungal pigment derived from 0.5 mg of dried fungal biomass was dissolved in 5 mL of KOH (1 M) for 48 h and autoclaved (20 min at 121 °C). Then, the mixture was centrifuged at 5000 rpm for 5 min, and the resulting supernatant was acidified with HCl (2M) to pH 2.5. Next, centrifugation at 5000 rpm for 5 min was performed to collect the precipitate, washed thrice with deionized water, dialyzed, and dried at 60 °C for 48 h. This pellet (fungal pigment) was kept at −20 °C until future use.

The obtained fungal pigment after purification (0.5 mg) was dissolved ©n 10 mL of KOH (1M) following the method described by [38]. The KOH (1M) solution was used as a reference blank. For the determination of extracted melanin concentration, synthetic melanin was prepared in 1 M KOH (1 M) at 2, 4, 6, 8, 8, 10 µg mL^−1^. A standard curve was performed at A 650, determining the relationship between logarithmic absorbance and wavelength.

### 2.3. Reactive Oxygen Species (ROS) and Lipid Peroxidation of Fungal Cell Membrane 

Fungal hyphae from the control, T1, and T2 were stained with 30 µM ROS staining probe (CM-H_2_DCFDA Thermofisher, Waltham, MA, USA) for 1 h at room temp under darkness. Control (+) samples were further incubated with 200 µM H_2_O_2_ and then centrifuged at 5000 rpm × 10 min and washed with 1× PBS to remove excess of the marker. Samples were visualized at excitation/emission wavelength 488 nm/530 nm [40]. The level of lipid peroxidation was assessed on fresh samples of roots from wheat by monitoring the thiobarbituric-acid-reacting substances (TBARS). For this, a malondialdehyde (MDA) concentration was used to detect cell damage. The absorbance was measured with a UV–vis spectrophotometer at 532, 600, and 440 nm to correct the interference generated by TBARS–sugar complexes [12]. The unit for lipid peroxidation was determined as equivalents of MDA contents (nmol g^−1^ FW).

### 2.4. Greenhouse Assay: Ggt Pathogenicity in Wheat Roots

To confirm Ggt pathogenicity, inoculum from each treatment was put on wheat plants (cv. Otto). Wheat seeds were sown in plastic pots containing 400 g of a sterile substrate (vermiculite, peat, sand; 1:1:1). The inoculum of Ggt (control, T1 and T2) was applied at 0.1% concentration. The seedlings were watered every 3 days, and Taylor and Foyd nutrient solution every 15 days [41]. The experimental design included plants inoculated with Ggt under normal conditions (control, c), Ggt under UV-A light (T1), and Ggt under changes of T° (T2). Each treatment was performed in quadruplicate. After 60 days, plants were carefully removed from the soil. The blackening root percentage was determined on a 0–100% scale (see Section 3.4).

### 2.5. Gaeumannomyces graminis var. tritici Quantification in Wheat Roots 

To quantify the Nº copies of the Ggt genome, total DNA from fresh samples obtained from wheat root tissue was extracted with a soil DNA Isolation Kit (dNeasy PowerSoil Pro, Hilden, Germany) according to the manufacturer’s instructions. The Ggt DNA was quantified by quantitative real-time PCR (qPCR). Standard curve was prepared in triplicate from 10-fold serial dilutions of Ggt genomic DNA from 0.8 to 8 × 10^−5^ ng µL^−1^ (obtained from a Ggt pure culture in PDA). A specific Ggt DNA fragment was amplified using GGT2F/GGT168R primer sets. The reaction mixture was carried out in a final volume of 12 µL, containing Brilliant II SYBR, Green QPCR master mix (Strata-gene, Agilent Technologies Company, Cedar Creek, TX, USA), 1 µL 1:10 Ggt DNA dilution (to determine standard curve) or 1 µL sample DNA, and 600 nM of each primer. The real-time PCR reaction was performed in an Applied Biosystems Step One™ Real-Time PCR System, (Foster City, CA, USA) under the following conditions: an initial denaturing step at 95 °C for 10 min and 35 cycles at 95 °C for 15 s, 58.4 °C for 20 s, and 72 °C for 40 s. To determine the copy number of Ggt DNA in the soil samples, the following formula was used [4,42]:DNA Ggt sample ngμL×1×10−9 m ggenome13
where 13 represents the number of copies of the amplified fragment in the Ggt genome, and:$$m\left(g\right)genome:\;\frac{Ggt\;genome\;weight\;\left(43,768,664\;bp\right)\times average\;MW\;double\;\left(660\frac{g}{mol}\right)}{n^{\underset{\text{-}}{{}^{\circ}}}\;avogadros}$$

### 2.6. Statistical Analysis 

Data were analyzed by one-way analysis of variance (ANOVA). Mean comparisons were made by Tukey’s test using SPSS software (IBM SPSS, Inc., Chicago, IL, USA). All experiments were performed in triplicate, and the values are shown as mean ± standard error (SE). The differences were significant, with a *p*-value less than or equal to 0.05.

## 3. Results

### 3.1. In Vitro Induction of Melanin Fungal Pigment

Different culture conditions showed morphological differences in Ggt growth, evidenced by differences in mycelia coloration. For example, in the control treatment (25 °C), mycelia showed a white pigmentation and cottony aspect, as shown in Figure 1A. Under UV-A light (T1), the fungi had a black pigmentation, and finally, under temperature change (T2), a dark brown coloration was observed in both treatments. T1 and T2 did not present a cottony mycelium.

### 3.2. Detection and Quantification of Melanin Fungal Pigment

Extraction of melanin pigments produced from Ggt was achieved from 0.5 mg of the dried fungal biomass. The purified pigment appeared as a dark brown color. Pigment extraction and purification yielded between 0.4 and 1.8 mg of the dried fungal biomass in the treatments (Figure 1B). In both the treatments, UV-A (T1) and temperature (T2), the changes were significantly (*p* ≤ 0.05) higher than the control. In this way, fungal inocula grown under T1 evidenced around 33% more melanin pigment production, whereas fungal inocula from T2 showed 13% melanin pigment production compared to the control.

### 3.3. Response of Fungal Cell Membrane to Melanin Activation

In order to detect the effect of melanin activation in the fungal cell membrane, reactive oxygen species (ROS) and lipid peroxidation were quantified. Thus, Ggt grown under the control, T1 and T2 treatments were stained with the ROS probe to assess fungal behavior to melanin activation. According to microscopic examination, the results revealed that both the T1 and T2 treatments showed higher green fluorescence related to a significant production of ROS as compared with the control) (Figure 2A). In this way, samples from the T1 treatments showed significant higher green fluorescence, 4-fold more than the control, and 30% more than those from T2 (Figure 2B).

### 3.4. Greenhouse Assay

To corroborate the effect of melanized fungi in plant colonization and infection, a greenhouse assay was carried out. According to the results, all treatments showed the presence of necrotic roots (determined by blackening roots). The blackening root percentage was determined on a 0–100% scale with a ruler while contrasted against a white background and was recorded on a scale from 0 to 4, where 0: no take-all; 1: 1% to 10%; 2: 11% to 30%; 3: 31% to 60%; and 4: 61% to 100% of the root system was affected (Figure 3A).

This infection determination was contrasted with quantifying Nº copies of the Ggt genome. In the control, around 20% of the plants were asymptomatic, whereas in T1 and T2, all plants were infected. Plants infected with inoculum from T1 (UV light) were the most infected, reaching about 50% with 61% to 100% of necrosis, T2 30%, and the control 20% (Figure 3B).

### 3.5. Gaeumannomyces graminis var. tritici Quantification in Wheat Roots

The results confirm that the number of Ggt copies determined by real-time PCR increased with the increase in melanin production by 4-fold (186,500 copies of Ggt genome) more and 3-fold (140,000 copies of Ggt genome) more in T1 and T2, respectively, compared to the control (Figure 4A). However, in the control the presence of Ggt (around 49,000 copies of Ggt genome) also was detected. These results were concordant with the number of blackening roots, where a greater number of blackened roots were evidenced in T1. In this way, a positive correlation between the quantification of the Ggt genome copies and mycelial melanin pigment production was observed (Figure 4B).

### 3.6. Oxidative Damage by Lipid Peroxidation (MDA Accumulation)

Considering that malondialdehyde (MDA) is an essential signal of stress, lipid peroxidation measured as TBARS (thiobarbituric acid reactive substance) was quantified in wheat roots from greenhouse assay. According to the results, the plants from T1 showed two-fold more MDA production (nmol g^−1^ FW) compared to the control and 40% more than T2 (Figure 4C), where a more imminent fungal damage with increasing melanin production for fungal inocula, mainly in T1, was evidenced.

## 4. Discussion

During routine laboratory work, the soil-borne pathogen *Gaeumannomyces graminis* (Ggt) conserved under ex situ conditions significantly decreased its pathogenicity. The strains showed a loss of dark mycelium (changing to white mycelium) and asymptomatic and/or low-infected plants. However, the mechanisms of Ggt pathogenicity to wheat roots are still not well understood, and most of the research on wheat-Ggt interactions has been focused on the biological characteristics of the disease [43], pathogen distribution [44], pathogen genetic diversity and the use of antagonists [6,45].

In the last decades, several reports evidenced the important role of melanin, a secondary metabolite that act as “fungal armor” to defend against environmental stresses such as UV light, oxidizing agents, and ionizing radiation [25,46,47]. In this way, it is reported that melanin pigments can perceive the energy of radiation (UV, visible light, and radiation) and convert it into useful reducing power for metabolic processes such as fungal pathogenicity [25]. For example, Chumley and Valent [48] reported that the ascomycete fungi *Magnaporthe oryzae* (causal agent of rice blast disease) were unable to infect rice when dark pigmentation was lost due to melanin deficiency, not allowing the generation of the necessary turgor pressure in the cell wall. Other cases relate to *Cryptococcus neoformans*, *Aspergillus fumigatus* and *Wangiella dermatitidis,* which have shown a reduction in pathogenicity in hosts [49,50,51]. The role of melanin in Ggt has been investigated, evidencing that melanin is necessary for the production of runner hyphae to infect roots [29,32]. Therefore, we hypothesized that the melanin production induced by UV-A light (T1) and temperature variation (30, 15 and 20 °C, T2) directly affect melanin recovery with a concomitant impact on fungal pathogenicity.

Our main finding evidenced that compared to the control, melanin was increased by 30% in Ggt mycelia exposed to UV-A light and by 16% in Ggt subjected to Tº variation, showing a dark pigmentation of mycelia. This response is produced because melanin is accumulated in fungal cell walls and plays vital roles in fungal survival in the presence of damaging events such as UV and temperature variation, among others [52,53]. However, fungal mycelia of both treatments have been shown to promote ROS production and fungal virulence. In other studies, an increase in TBARS content is also associated with the mycelium of *Pleurotus eryngii* var. *tuoliensis* using heat as a stress-inducing factor [54]. In this way, non-optimal conditions of fungal growth can unveil novel virulence genes that are not evident in optimal conditions, i.e., salt stress and temperature variation [55,56]. On the other hand, melanin concentration was directly related to Ggt pathogenicity, considering blackening roots and Nº copies of the Ggt genome to 4-fold more and 3-fold more in T1 and T2, respectively, compared to the control. Increased pathogenicity also was evidenced by the increase in TBARS production, which reveals a significant membrane damage of the roots.

Fouly et al. [57] reported that aggressive melanized Ggt could develop ectotrophic melanized hypha able to colonize the epidermal and cortical tissues, resulting in the subsequent invasion and colonization of the endodermis and stele tissues in vascular occlusion. Thus, while not all pathogenic fungi are melanotic, there is a large class of potentially invasive fungi that have in common the production of melanin. For example, melanin is indispensable for *Venturia inaequalis* pathogenicity; nevertheless, its formation is restricted to structures and growth stages permanently exposed to the environment, whereas melanin-free structures within the plant are protected by the cuticle [34].

Finally, our study confirms that melanin production could be induced by inductor factors such as UV-A light (T1) and temperature variations (30, 15 and 20 °C, T2) with a concomitant effect in fungal pathogenicity. However, it is necessary to expand the knowledge about the pathogenic genes involved in melanin activation. We hope that the information in this article will be useful to solutions to be applied in the laboratory to recovery fungal pathogenicity.

## 5. Conclusions

The present study revealed that melanin is indispensable for *Gaeumannomyces graminis* (Ggt) to be a successful pathogen. This was evidenced by melanin induction using UVA-A and Tº changes in low-melanized fungi preserved under an oil overlay. UVA-A was the main inductor factor of melanin in Ggt, producing major infections in inoculated plants verified by blackening roots, Nº copies of the Ggt genome, and major membrane damage (TBARS). In this context, we hope that the information in this manuscript will be useful for application in the laboratory to recover Ggt pathogenicity. 

## Figures and Tables

**Figure 1 jof-09-00350-f001:**
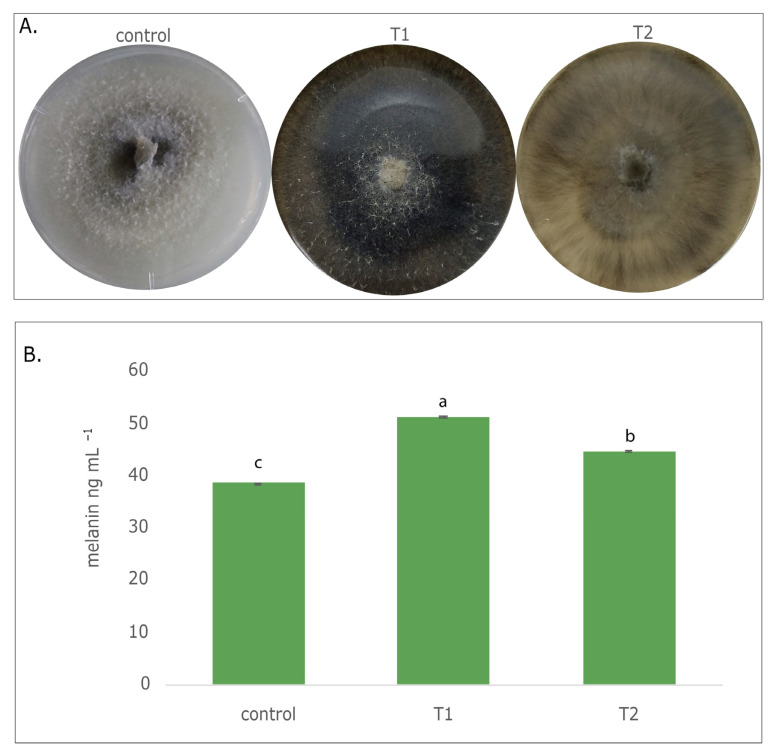
(**A**) Melanin fungal pigment production, (c) control, T1: Ultraviolet light (UV-A 320–400 nm) as inductor factor, T2: Temperature (30, 15 and 20 °C) as inductor factor. (**B**) Melanin concentrations (ng mL^−1^). Different letters denote significant difference (*p* ≤ 0.05).

**Figure 2 jof-09-00350-f002:**
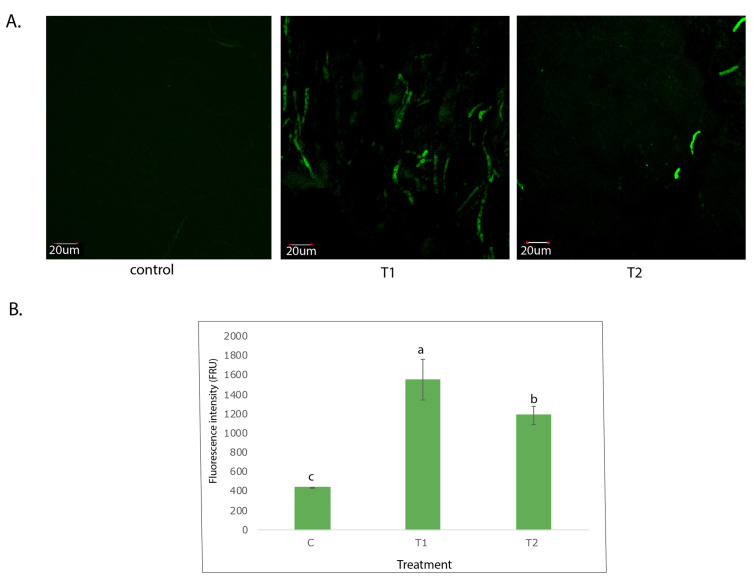
(**A**) Detection of reactive oxygen species (ROS) within *Gaeumannomyces graminis* hyphae staining with the oxidatively active fluorescent dye 2,7-dichlorodihydrofluorescein diacetate (H_2_DCFDA, Molecular Probes). (**B**) Fluorescence intensity expressed as FRU (Fluorescent Relative Unit). Different letters denote a significant difference (*p* < 0.05; Tukey test) between treatments. (c) control, T1: Stress condition under UV-A stress, T2: Stress condition due to temperature change (30, 15, and 20 °C).

**Figure 3 jof-09-00350-f003:**
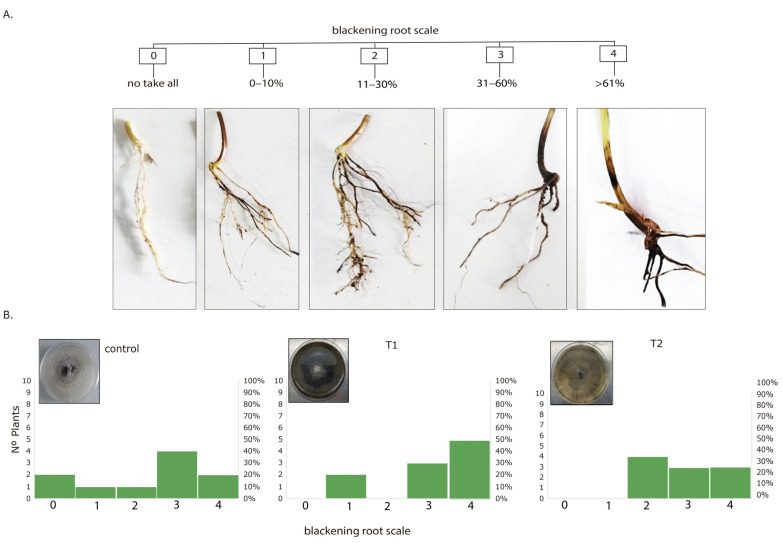
(**A**) Infection scale. 0: no take-all; 1: 1% to 10%; 2: 11% to 30%; 3: 31% to 60%; and 4: 61% to 100% of blackening roots, (**B**) Nº plants infected with take-all (blackening roots) according to infection scale after 60 days.

**Figure 4 jof-09-00350-f004:**
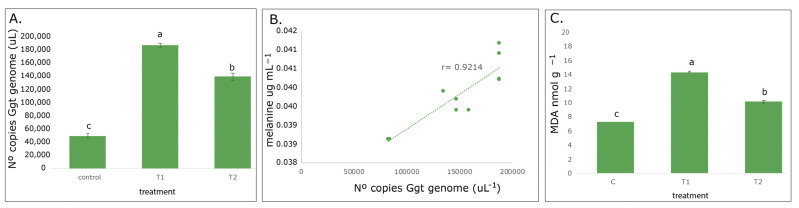
(**A**) Number of Ggt genome copies in infected roots 60 days after inoculation with 1% of Ggt. (**B**) Correlation between melanin concentration and Ggt genome copy number. (**C**) Lipid peroxidation measured as TBARS (thiobarbituric acid reactive substance) roots. (c) control, inducer factors T1: UV-A, T2: temperature variation (30, 15 and 20 °C). Different letters denote significant difference (*p* ≤ 0.05).

## Data Availability

Not applicable.

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
