# Peer review of "Melanin Induction Restores the Pathogenicity of Gaeumannomyces graminis var. tritici in Wheat Plants"

_jof, 2023, doi:10.3390/jof9030350_

Round 1

Reviewer 1 Report

This paper does not clarify whether the strain used is white strain after long-term transfer, or normal strain. If it is the former (white strain), a normal strain should be set as a control. In addition, cryopreservation is not expensive or complicated, and a refrigerator is enough for that preservation. For the rice blast fungus, its Latin scientific name Magnaporthe grisea is now rarely used, and Magnaporthe oryzae or Pyricularia oryzae is the canonical name.

Author Response

Reviewer 1.

We thanks to Reviewer 1 for your important contribution in the improvement of the present manuscript.

This paper does not clarify whether the strain used is white strain after long-term transfer, or normal strain. If it is the former (white strain), a normal strain should be set as a control.

Author response: We select a white strain (or low melanized strains) as a control  because our hypothesis was that ultraviolet light and temperature promote the formation of melanin and consequent influence on wheat plant pathogenicity. A normal strain will not respond to question that we need answered because also is melanized. 

In addition, cryopreservation is not expensive or complicated, and a refrigerator is enough for that preservation.

Author response: we added the following information: For example, for ex-situ conservation by cryopreservation the tubes should be frozen in ultracold-freezer from -70 °C to -196 °C and periodically inoculated to evaluate fungal viability, purity, and phenotypic characteristics . The method of preservation by lyophilisation consists of drying at -50 °C and vacuum sublimation. However, not all fungi survive the process conditions and storage in liquid nitrogen is more expensive than lyophilization and liquid nitrogen should be replenished every few days.

  1. García-Martínez, J.; López Lacomba, D.; Castaño Pascual, A. Evaluation of a Method for Long-Term Cryopreservation of Fungal Strains. Biobank. 2018, 16, 128–137, doi:10.1089/bio.2017.0101.
  2. Ayala-Zermeño, M.A.; Gallou, A.; Berlanga-Padilla, A.M.; Andrade-Michel, G.Y.; Rodríguez-Rodríguez, J.C.; Arredondo-Bernal, H.C.; Montesinos-Matías, R. Viability, purity, and genetic stability of entomopathogenic fungi species using different preservation methods. Fungal Biol. 2017, 121, 920–928, doi:10.1016/j.funbio.2017.07.007.
  3. Al-Bedak, O.A.; Sayed, R.M.; Hassan, S.H.A. A new low-cost method for long-term preservation of filamentous fungi. Agric. Biotechnol. 2019, 22, 101417, doi:10.1016/j.bcab.2019.101417.

For the rice blast fungus, its Latin scientific name Magnaporthe grisea is now rarely used, and Magnaporthe oryzae or Pyricularia oryzae is the canonical name.

Author response: We replaced Magnaporthe grisea  by Magnaporthe oryzae

Reviewer 2 Report

Dear Authors, 

I have several comments regarding the submitted manuscript. Please find them hereinafter: 

- I recommend a general check of the English language of the whole manuscript. 

- explain better the concepts. Check the updated file for the parts that I consider poorly or wrongly explained. 

-Abstract:

Line26. Explain better the sentence. Real time is not a direct measure of pathogenicity. 

- the introduction can be enriched with more literature, also considering more fungal species where the melanin has a role in pathogenicity as done in the discussion. 

- M&M 2.4 Include how many plants have been used for each treatment

- formula on page 4 has to be formatted

-Results.

3.1 & 3.2 It is not clear if the different amount of melanin among the treatment was evaluated starting from the same amount of biomass. I assume so but this has to be written in the text.

- line 201. Wrong numbering of the paragraph 

3.4 results about the qPCR are missing . The paragraph is poorly written. 

-Figure 3B. Put the bars of the three graphs in the same order. In this way     the graphs are confusing and not easy to consult. 

- the conclusion are poorly written. 

Best Regards. 

Author Response

Reviewer 2.

1.- I recommend a general check of the English language of the whole manuscript. 

Author response: English was revised in all document

- explain better the concepts. Check the updated file for the parts that I consider poorly or wrongly explained. 

-Abstract:

Line26. Explain better the sentence. Real time is not a direct measure of pathogenicity. 

Author response: We rephrase the sentence “Fungal pathogenicity was evaluated by determination of blackening roots and Ggt was quantified by Real Time PCR in inoculated wheat plants”

- the introduction can be enriched with more literature, also considering more fungal species where the melanin has a role in pathogenicity as done in the discussion. 

Author response: We added the information, lines 172-174

- M&M 2.4 Include how many plants have been used for each treatment

Author response: We added the information line …

- formula on page 4 has to be formatted

Author response: Formula was inserted as a formula format

-Results.

3.1 & 3.2 It is not clear if the different amount of melanin among the treatment was evaluated starting from the same amount of biomass. I assume so but this has to be written in the text.

Author response:  we used 0.5 mg of dried fungal biomass (information was added in line 279)

- line 201. Wrong numbering of the paragraph 

Author response:  numbering of the paragraph was checked in all document

3.4 results about the qPCR are missing . The paragraph is poorly written. 

Author response:  we rewrite the paragraph

-Figure 3B. Put the bars of the three graphs in the same order. In this way     the graphs are confusing and not easy to consult. 

Author response:  Figure was modified according your recommendation

- the conclusion are poorly written. 

 Author response:  we improve conclusions

Reviewer 3 Report

The importance of melanin at disease resintance is not very new, but the authors continue reserach and accumulate results in this field, I agree this paper will be pblished in this Journal. 

I added some comments, mainly about the Figure and statisitics. Please see attached file.

Author Response

We thank to Reviewer 3, all your recommendations were included in the new version of the manuscript

Round 2

Reviewer 1 Report

DrIed spores or hyphae of fungi can be stored under -20°C for long periods of time, so, in my opinion, ultra-low temperatures are not always necessary for fungal preservation.